# Overcoming Aging-Associated Poor Influenza Vaccine Responses with CpG 1018 Adjuvant

**DOI:** 10.3390/vaccines10111894

**Published:** 2022-11-10

**Authors:** Xinliang Kang, Yibo Li, Yiwen Zhao, Xinyuan Chen

**Affiliations:** Biomedical and Pharmaceutical Sciences, College of Pharmacy, University of Rhode Island, 7 Greenhouse Road, Pharmacy Building, Room 480, Kingston, RI 02881, USA

**Keywords:** CpG 1018, CpG, aged mice, influenza vaccine, immunosenescence, aging

## Abstract

Aging is associated with diminished immune system function, which renders old people vulnerable to influenza infection and also less responsive to influenza vaccination. This study explored whether the CpG 1018 adjuvant was effective in enhancing influenza vaccine efficacy in aged mice equivalent to human beings in their late 50s to early 60s. Using the influenza pandemic 2009 H1N1 (pdm09) vaccine as a model, we found that the CpG 1018 adjuvant could significantly enhance the pdm09 vaccine-induced serum antibody titer, while the pdm09 vaccine alone failed to elicit significant antibody titer. In contrast, the pdm09 vaccine alone elicited significant antibody titer in young adult mice. Antibody subtype analysis found that the pdm09 vaccine alone elicited Th2-biased antibody responses in young adult mice, while incorporation of the CpG 1018 adjuvant promoted the elicitation of potent Th1-biased antibody responses in aged mice. The pdm09 vaccine alone was further found to induce significant expansion of Th2 cells in young adult mice, while incorporation of the CpG 1018 adjuvant stimulated significant expansion of Th1 cells in aged mice. The CpG 1018 adjuvant also stimulated vaccine-specific cytotoxic T lymphocytes in aged mice. The pdm09 vaccine in the presence of CpG 1018 elicited significant protection against lethal viral challenges, while the pdm09 vaccine alone failed to confer significant protection in young adult or aged mice. Our study provided strong evidence to support the high effectiveness of the CpG 1018 adjuvant to boost influenza vaccination in aged mouse models.

## 1. Introduction

Influenza is a highly contagious viral infectious disease and can cause severe illnesses or even deaths [1,2]. Aging has been a well-known risk factor for severe diseases after influenza infection [3,4]. According to the Center for Disease Control and Prevention (CDC), approximately 90% of influenza-associated deaths and 50–70% of influenza-associated hospitalizations occurred in people aged 65 years and older. The hospitalization rate of people aged 65 years and older was 6–7-fold higher than their young counterparts (5–64 years) after contracting influenza, according to an epidemiological study [5]. The weakened immune system function has been the major reason of severe diseases associated with influenza in old people. Aging was found to impair both innate and adaptive immune responses against influenza infection. Excessive and persistent neutrophil accumulation, reduced phagocytosis of alveolar macrophages, reduced dendritic cell (DC) number and migration have been found to be associated with aging, which caused immunopathological damage of the lung and poor stimulation of influenza-specific T and B cell responses [6]. Aging also reduces the number and repertoire of CD4^+^ and CD8^+^ T cells, leading to weakened helper T cell (Th) and cytotoxic T lymphocyte (CTL) responses [7]. Impaired Th responses also lead to reduced B cell and humoral immune responses in the elderly [8]. Other medical conditions, especially bacterial co-infection, could worsen symptoms of influenza infection in the elderly [9].

Age-associated immunosenescence not only reduces the antiviral immune responses after infection but also weakens influenza vaccine-induced protective immune responses in the elderly. A systematic review and meta-analysis found lower influenza vaccine effectiveness in the elderly (30–44%) than in younger adults (45–58%) between the 2010–11 and 2014–15 flu seasons [10]. In recognizing such a difference, better vaccines have been explored and approved for the elderly that include the high-dose influenza vaccine and MF59-adjuvanted influenza vaccine. The high-dose influenza vaccine contains four times more hemagglutinin antigen (HA) than the standard-dose influenza vaccine, while the MF59-adjuvanted vaccine uses MF59 to enhance influenza vaccine efficacy in the elderly. One study found that the high-dose influenza vaccine more significantly prevented influenza or pneumonia-caused hospitalizations and emergency room visits than the standard-dose influenza vaccine in the elderly during the 2012–2018 flu seasons in the United States [11]. A second study found that the MF59-adjuvanted influenza vaccine was similarly effective to the high-dose influenza vaccine and more effective than the standard-dose influenza vaccine to prevent influenza-related hospitalizations and emergency room visits [12]. The high-dose influenza vaccine enhances the magnitude and is not expected to modify immune response types, while the MF59-adjuvanted vaccine enhances the magnitude and also expands the breadths of vaccine-induced antibody responses. Yet, MF59 mainly induces Th2-biased immune responses with little ability to induce CTL responses [13], which play a key role in eliminating virus-infected cells and promoting recovery [14].

CpG 1018 is an oligonucleotide-based vaccine adjuvant composed of unmethylated CpG capable of activating the toll-like receptor (TLR) 9 [15]. CpG 1018 was recently approved to boost hepatitis B vaccine efficacy [16]. The CpG 1018-adjuvanted hepatitis B vaccine was found to elicit more potent immune responses than the Alum-adjuvanted hepatitis B vaccine [15]. Different from species-specific CpG molecules, CpG 1018 is broadly effective in mice, non-human primates, and human beings and can be used to support vaccine development without the need to change CpG molecules in preclinical and clinical studies [15]. The use of different CpG molecules may complicate data interpretation due to their potentially different adjuvant effects. CpG is a Th1 adjuvant and can assist protein antigens to elicit potent CTL responses [17,18]. Our recent studies found that the CpG 1018 adjuvant was more effective than the MF59-mimetic AddaVax adjuvant in enhancing influenza nucleoprotein-induced immune responses and protection in murine models (manuscript under review). 

This study used the influenza pandemic 2009 H1N1 (pdm09) vaccine as an example to explore whether CpG 1018 was effective in enhancing its humoral and cellular immune responses and protection in an aged mouse model. 

## 2. Materials and Methods

### 2.1. Reagents

Recombinant HA (rHA) of Influenza A/California/07/2009 (H1N1) (FR-559) was obtained from International Reagent Resource (IRR, Manassas, VA, USA). Monovalent 2009 H1N1 influenza (pdm09) vaccine (NR-20083) was obtained from BEI Resources (Manassas, VA, USA). Fluorescence-conjugated antibodies used in immunostaining and flow cytometry were purchased from Biolegends (San Diego, CA, USA). Horseradish peroxidase (HRP)-conjugated anti-mouse IgG secondary antibodies (45000679) was obtained from Fisher Scientific (Waltham, MA, USA). CpG 1018 was synthesized by Trilink Biotechnologies (San Diego, CA, USA). Chicken red blood cells (RBCs) were obtained from Charles River (Wilmington, MA, USA). 

### 2.2. Mice

Male young adult (6–8 weeks old) and aged C57BL/6 mice (76 weeks old) were purchased from the Jackson Laboratory (Bar Harbor, ME, USA). Aged mice were used when they reached 78 weeks old. Mice were housed in the animal facilities of the University of Rhode Island (URI). Influenza viral challenge studies were conducted in the animal biosafety level 2 (ABSL2) facility of URI. All procedures were approved by the Institutional Animal Care and Use Committee (IACUC) of URI with the approval number AN1516-004.

### 2.3. Immunization

Intradermal (ID) immunization was used in our study to elicit better immune responses than traditional intramuscular immunization [19]. For accurate ID immunization, hair on the lateral dorsal skin of mice was removed one day before the experiment as shown in our recent studies [20,21]. Next day, aged mice were intradermally immunized with 0.5 µg pdm09 vaccine (HA equivalent) alone or in the presence of 40 µg CpG 1018 or left non-immunized. Young adult mice were intradermally immunized with 0.5 µg pdm09 vaccine or left non-immunized to serve as comparisons. 

### 2.4. Hemagglutination Inhibition (HI) Titer

Serum HI titer was measured as in our previous reports [22,23]. In brief, serum samples were treated with receptor-destroying enzyme II (Hardy Diagnostics, Santa Maria, CA, USA), heat inactivated, and further adsorbed with chicken RBCs. Serum samples were then subjected to a twofold serial dilution and incubation with four hemagglutinating units of pdm09 viruses (A/California/07/2009). Chicken RBCs were added, and the agglutination patterns were read to determine the HI titer. 

### 2.5. ELISA Antibody Titer

Serum antibody titer was measured by enzyme-linked immunosorbent assay (ELISA) as in our prior reports [20,21,23]. In brief, serum samples were subjected to twofold serial dilutions and added to 96-well ELISA plates pre-coated with 1 μg/mL rHA. After 90 min incubation and washing, HRP-conjugated anti-mouse IgG secondary antibodies were added. After 1 h incubation and washing, TMB substrates were added. The optical absorbance (OD_450nm_) was read in a microplate reader after the addition of 3M H_2_SO_4_. For detection of subtype antibody titer, HRP-conjugated anti-mouse IgG1 and IgG2c secondary antibodies were used [20,21,23].

### 2.6. Cell-Mediated Immune Responses

Whole blood (~50 μL) was collected into heparinized tubes and centrifuged at 1300 rpm for 5 min. RBCs were lysed in ACK lysis buffer. Peripheral blood mononuclear cells (PBMCs) were seeded into 96-well plates and stimulated with 10 μg/mL rHA overnight in the presence of 4 μg/mL anti-CD28 antibodies (37.51). Brefeldin A (420601, Biolegend) was added and PBMCs were harvested 5 h later. PBMCs were stained with fluorescence-conjugated anti-CD4 (GK1.5) and anti-CD8 antibodies (53–6.7). Cells were then fixed, permeabilized, and stained with fluorescence-conjugated anti-IFNγ (XMG1.2) and anti-IL4 antibodies (11B11). Cells were subjected to flow cytometry in BD FACSVerse and data were analyzed by FlowJo software. 

### 2.7. Lethal Viral Challenge

Mice were intranasally inoculated with 8 × LD_50_ of mouse-adapted pdm09 viruses under light anesthesia as in our previous reports [21,23]. Body weight and survival were monitored daily for 14 days. Mice with body-weight loss more than 20% were euthanized and regarded as dead. 

### 2.8. Statistics

Values were expressed as mean ± SEM (standard error of the mean). One-way analysis of variance (ANOVA) with Tukey’s multiple comparison test was used to compare differences for more than two groups, except otherwise specified. Log-rank (Mantel–Cox) test with Bonferroni correction was used to compare differences of survival between groups. *p*-value was calculated by PRISM software (GraphPad, San Diego, CA, USA) and considered significant if it was <0.05.

## 3. Results

### 3.1. CpG 1018 Enhances HI Titer

Old mice (78 weeks) were subjected to ID immunization of the pdm09 vaccine (0.5 µg) alone or in the presence of 40 µg of the CpG 1018 adjuvant or left non-immunized. Young adult mice (6–8 weeks) mice were subjected to ID immunization of the pdm09 vaccine of the same dose or left non-immunized to serve as controls. Experimental procedures and timelines were illustrated in Figure 1. The serum HI titer was measured 3 weeks after immunization. As shown in Figure 2, the pdm09 vaccine alone failed to significantly increase the serum HI titer in aged mice, while the pdm09 vaccine alone significantly increased the serum HI titer in young adult mice. Incorporation of the CpG 1018 adjuvant in pdm09 vaccination significantly increased the HI titer as compared to the vaccine alone or the non-immunized group in aged mice (Figure 2). Furthermore, the pdm09 vaccine in the presence of the CpG 1018 adjuvant also elicited significantly higher HI titer in aged mice than the pdm09 vaccine alone in young adult mice (Figure 2). This study indicated weakened immune responses of aged mice to pdm09 vaccination and CpG 1018 was effective to overcome the weakened immune responses and elicit even more potent immune responses in aged mice than the vaccine alone in young adult mice. 

### 3.2. CpG 1018 Enhances ELISA Antibody Titer

Besides the HI titer, we also measured the pdm09 vaccine-induced ELISA IgG titer. As shown in Figure 3, the vaccine alone in aged mice failed to induce a significant anti-HA IgG titer, while the vaccine alone in young adult mice induced a significant IgG titer. The pdm09 vaccine in the presence of CpG 1018 induced a significant anti-HA IgG titer in aged mice (Figure 3). The serum anti-HA IgG titer in the vaccine/CpG 1018 group in aged mice was also higher than that in the vaccine alone group in young adult mice despite the lack of significant difference.

Besides the total IgG, the subtype IgG1 and IgG2c antibody responses were also explored and compared between groups. As shown in Figure 4A, the vaccine alone significantly increased HA-specific IgG1 production in young adult mice but not in aged mice. Incorporation of CpG 1018 also failed to significantly increase the serum IgG1 antibody titer in aged mice (Figure 4A). In contrast, incorporation of CpG 1018 vigorously enhanced HA-specific IgG2c antibody production in aged mice, while the vaccine alone failed to significantly enhance HA-specific IgG2c antibody production in young adult or aged mice (Figure 4B). This study indicated the ability of CpG 1018 to elicit potent Th1-biased antibody responses in aged mice. 

### 3.3. CpG 1018 Induces Th1 Cells and CTLs

The ability of CpG 1018 to directly activate IFNγ-secreting CD4^+^ T or Th1 cells and IL4-secreting CD4^+^ T or Th2 cells as well as IFNγ-secreting CD8^+^ T cells or CTLs and IL4-secreting CD8^+^ T cells was then explored in PBMCs one week after immunization. The vaccine alone most significantly expanded Th2 cells in young adult mice and such an expansion was not clear in aged mice (Figure 5). The pdm09 vaccine in the presence of CpG 1018 expanded Th1 cells and CTLs in aged mice, while the expansion of Th2 cells or IL4-secreting CD8^+^ T cells was not clear (Figure 5). 

Quantitative analysis found that the vaccine alone elicited significantly higher frequencies of Th2 cells in young adult but not aged mice (Figure 6A). The vaccine alone failed to increase frequencies of Th1 cells in young adult or aged mice (Figure 6B). The vaccine in the presence of CpG 1018 reduced frequencies of Th2 cells when compared with the vaccine alone in aged mice (Figure 6A). However, such a difference did not reach a statistically significant level. Instead, the vaccine in the presence of CpG 1018 induced significantly higher frequencies of Th1 cells when compared to the vaccine alone or non-immunized group in aged mice (Figure 6B). The frequencies of Th1 cells in the vaccine/CpG 1018 group in aged mice were also significantly higher than that of the vaccine alone in young adult mice (Figure 6B). Pdm09 vaccination in the presence of CpG 1018 significantly reduced frequencies of IL4-secreting CD8^+^ T cells as compared to the vaccine alone or non-immunized group (Figure 6C). In contrast, pdm09 vaccination in the presence of CpG 1018 significantly increased frequencies of CTLs as compared to the vaccine alone (Figure 6D). The vaccine alone failed to significantly increase CTLs or IL4-secreting CD8^+^ T cells in young adult or aged mice (Figure 6C,D). These results indicated that the pdm09 vaccination in the presence of CpG 1018 could induce potent Th1 and CTL responses in aged mice, while the pdm09 vaccine alone could induce Th2 cells in young adult but not aged mice. 

### 3.4. CpG 1018 Enhances Vaccine-Induced Protection

Mice were challenged with a lethal dose of pdm09 viruses to explore protection. As shown in Figure 7A, vaccine alone failed to elicit significant protection against body weight loss in aged mice as evidenced by the similar trend of body-weight loss between non-immunized and vaccine alone groups. In contrast, the vaccine in the presence of CpG 1018 induced significant protection against body-weight loss (Figure 7A). Significantly less body-weight loss was observed in the vaccine/CpG 1018 group when compared to the vaccine alone group in aged mice on day 5–7 post infection (Figure 7A). Mice in the vaccine/CpG 1018 group lost a maximum of ~12% body weight on day 6 and recovered to its original body weight on day 12 (Figure 7A). In contrast, mice of all other groups either died or reached the humane euthanasia point between day 5–8 after challenge. Three of the seven aged mice survived the challenge in the vaccine/CpG 1018 group (Figure 7B). The survival rate in the vaccine/CpG 1018 group was significantly higher than that in the vaccine alone or non-immunized group in aged mice (Figure 7B). The vaccine alone in young adult mice reduced the body-weight loss especially on day 4 and 5 after the challenge (Table 1) and yet failed to prevent lethality.

## 4. Discussion

Our study found CpG 1018 was highly effective in enhancing pdm09 vaccine-induced immune responses and protection against lethal viral challenges in aged mice. The pdm09 vaccine in the presence of the CpG 1018 adjuvant elicited potent antibody responses in aged mice. The pdm09 vaccine with CpG 1018 but not the pdm09 vaccine alone significantly increased the serum HI titer and anti-HA IgG titer in aged mice. In contrast, the pdm09 vaccine alone could significantly increase the serum HI titer and anti-HA IgG titer in young adult mice. These results indicated lower influenza vaccine responses in aged mice, which could be overcome by the incorporation of the CpG 1018 adjuvant. One prior report found that the CpG adjuvant was only effective to enhance influenza vaccine efficacy in young adult but not aged mice [24]. The lack of efficacy of the CpG adjuvant in aged mice in the prior study remained to be explored but might be caused by the different CpG doses and immunization routes used. The prior study used a lower CpG dose (10 µg) and intramuscular immunization. Our study used a higher CpG dose (40 µg) and ID immunization. ID immunization has been well known to induce more potent immune responses than intramuscular immunization due to the richness of antigen-presenting cells in the skin rather than the muscle [19,25,26]. ID delivery of influenza vaccines in the elderly also elicits better immune responses than intramuscular vaccine delivery [27,28]. Higher CpG doses and more immunogenic ID immunization might result in better immune responses observed in our study. CpG classes were also different between the two studies. It remains to be explored whether class B CpG 1018 used in our study is more potent than class C CpG 2395 used in the prior study in boosting influenza vaccination [29].

Subtype antibody analysis found that the pdm09 vaccine mainly induced anti-HA IgG1 antibody responses in young adult mice and failed to elicit significant anti-HA IgG1 or IgG2c antibody responses in aged mice. The pdm09 vaccine in the presence of CpG 1018 was found to elicit significant anti-HA IgG2c but not IgG1 antibody responses in aged mice. These results indicated that the pdm09 vaccine alone mainly induced Th2-biased antibody responses in young adult mice, while the pdm09 vaccine in the presence of CpG 1018 mainly induced Th1-biased antibody responses in aged mice. The conversion from Th2 to Th1-biased antibody responses was expected to be due to the impact of the CpG 1018 adjuvant. 

The pdm09 vaccine was found to induce significant Th2 but not Th1 cells in young adult mice, while it failed to elicit significant Th2 cells in aged mice. These results indicated weakened Th responses in aged mice. Incorporation of the CpG 1018 adjuvant was found to significantly increase Th1 cells and suppress Th2 cells in aged mice, indicating that CpG 1018 was a Th1 adjuvant, in line with its ability to induce Th1-biased antibody responses. Besides Th1 cells, CpG 1018 also significantly increased CTLs and suppressed IL4-secreting CD8^+^ T cells. Our studies indicated that the CpG 1018 adjuvant was effective in enhancing influenza vaccine-induced Th1 and CTL responses in aged mice.

The pdm09 vaccine in the presence of CpG 1018 but not the pdm09 vaccine alone conferred significant protection in aged mice. Interestingly, the pdm09 vaccine alone failed to elicit significant protection in young adult mice, which might be due to the relatively low vaccine dose used (0.5 µg). In support, our previous studies found that 0.3 µg pdm09 vaccine failed to elicit significant protection in the same species of mice [22,23]. The induction of significant protection by the pdm09 vaccine in the presence of CpG 1018 hinted at the high effectiveness of this adjuvant to boost influenza vaccination in aged mice. Due to the induction of significant antibody and CTL responses in the presence of CpG 1018, we believe both humoral and cell-mediated immune responses contributed to the observed protection, although their relative contribution remains to be studied. Due to the induction of potent CTL responses and the conserved CTL epitopes of HA [14], influenza vaccination in the presence of the CpG 1018 adjuvant might elicit cross-protective immunity. The relative potency of CpG 1018 to the AddaVax adjuvant in enhancing influenza vaccine efficacy in aged mice can also be directly compared in future studies.

To our knowledge, this is the first study to show the high effectiveness of a clinically approved CpG 1018 adjuvant to enhance influenza vaccine efficacy in aged mice with diminished immune system functions. CpG 1018 has unique advantages that enhance influenza vaccine efficacy in the elderly. CpG 1018 is a 22-mer nucleotide-based adjuvant and can be easily synthesized in large scales. CpG 1018 is also expected to have a good stability during long-term storage. In contrast, other approved adjuvants (Alum, MF59, AS03, AS01, AS04) are all nano-formulations (e.g., nanocrystals, nano-emulsions, liposomes) [16,30], which have a risk of self-aggregation or phase separation during long-term storage. The strong enhancement of both arms of immunity, easy synthesis, and good stability support further investigation of the CpG 1018 adjuvant to boost influenza vaccination in the elderly.

## 5. Conclusions

The proportion of the world’s population over 60 years is expected to double between 2015–2050 according to the World Health Organization. Due to the weakened immune system function, more immunogenic influenza vaccines are highly demanded to protect the growing aging population. Our current study proved that CpG 1018 is a highly effective adjuvant that enhances influenza vaccine-induced immune responses and protection in aged mice and supports further investigation of the CpG 1018 adjuvant to overcome aging-associated poor influenza vaccine responses in the elderly.

## Figures and Tables

**Figure 1 vaccines-10-01894-f001:**
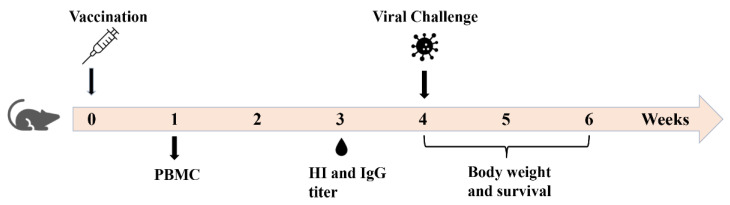
Timeline of experimental procedures.

**Figure 2 vaccines-10-01894-f002:**
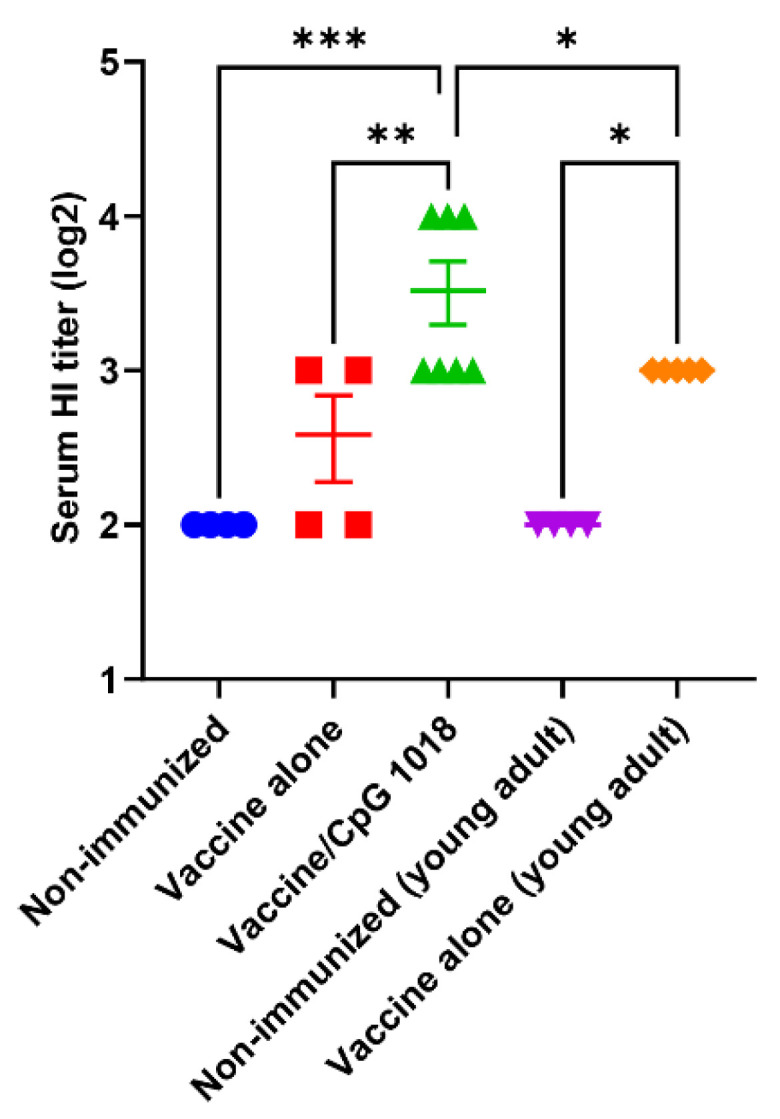
CpG 1018 increases pdm09 vaccine-induced HI titer. Old mice (78 weeks) were intradermally immunized with 0.5 µg pdm09 vaccine alone (

) or in the presence of CpG 1018 adjuvant (40 µg, 

) or left non-immunized (

). Young adult mice (6–8 weeks) were intradermally immunized with the same amount of pdm09 vaccine (

) or left non-immunized (

). Serum HI titer was measured 3 weeks after immunization. *n* = 4–7. One-way ANOVA with Fisher’s LSD test was used to compare differences between groups. *, *p* < 0.05; **, *p* < 0.01; ***, *p* < 0.001.

**Figure 3 vaccines-10-01894-f003:**
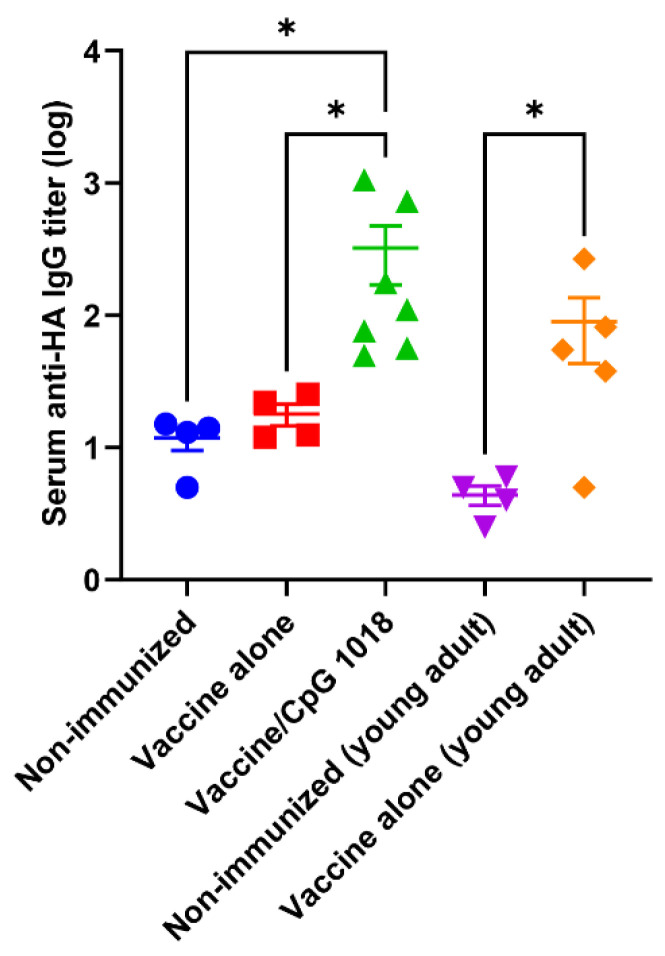
CpG 1018 enhances ELISA IgG titer. Old mice (78 weeks) were intradermally immunized with 0.5 µg pdm09 vaccine alone (

) or in the presence of CpG 1018 adjuvant (40 µg, 

) or left non-immunized (

). Young adult mice (6–8 weeks) were intradermally immunized with the same amount of pdm09 vaccine (

) or left non-immunized (

). Serum anti-HA IgG titer was measured 3 weeks after immunization by coating ELISA plate with rHA of pdm09 virus. *n* = 4–7. One-way ANOVA with Kruskal–Wallis multiple comparison test was used to compare differences between groups. *, *p* < 0.05.

**Figure 4 vaccines-10-01894-f004:**
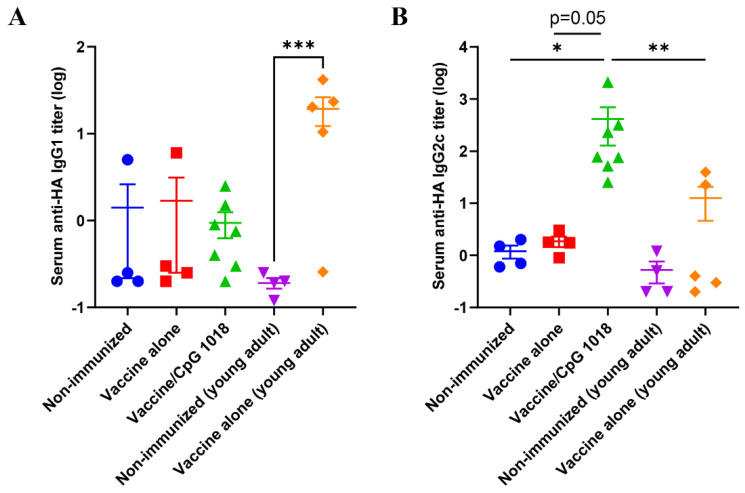
CpG 1018 induces strong IgG2c and weak IgG1 antibody responses. Old mice (78 weeks) were intradermally immunized with 0.5 µg pdm09 vaccine alone (

) or in the presence of CpG 1018 adjuvant (40 µg, 

) or left non-immunized (

). Young adult mice (6–8 weeks) were intradermally immunized with the same amount of pdm09 vaccine (

) or left non-immunized (

). Serum anti−HA IgG1 (**A**) and IgG2c antibody titers (**B**) were measured 3 weeks after immunization by coating ELISA plate with rHA of pdm09 virus. *n* = 4–7. One-way ANOVA with Kruskal–Wallis multiple comparison test was used to compare differences between groups. *, *p* < 0.05; **, *p* < 0.01; ***, *p* < 0.001.

**Figure 5 vaccines-10-01894-f005:**
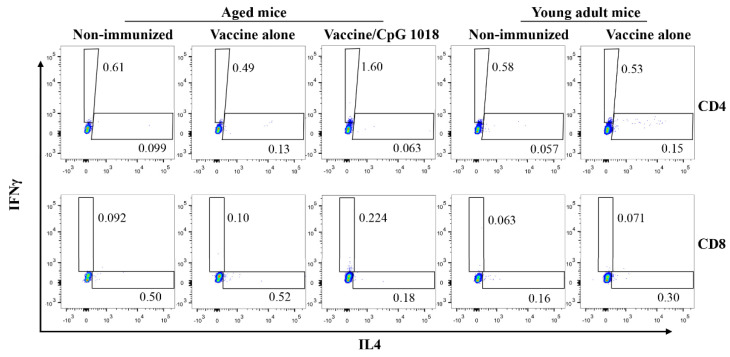
CpG 1018 promotes the induction of Th1 cells and CTLs in PBMCs. PMBCs were collected one week after immunization and stimulated with rHA overnight followed by immune staining and flow cytometry analysis. Cells were first gated based on FSC and SSC and then on CD4 and CD8 expression (Appendix A). IFNγ and IL4−secreting cells were then analyzed in CD4^+^ and CD8^+^ T cells. Representative dot plots were shown.

**Figure 6 vaccines-10-01894-f006:**
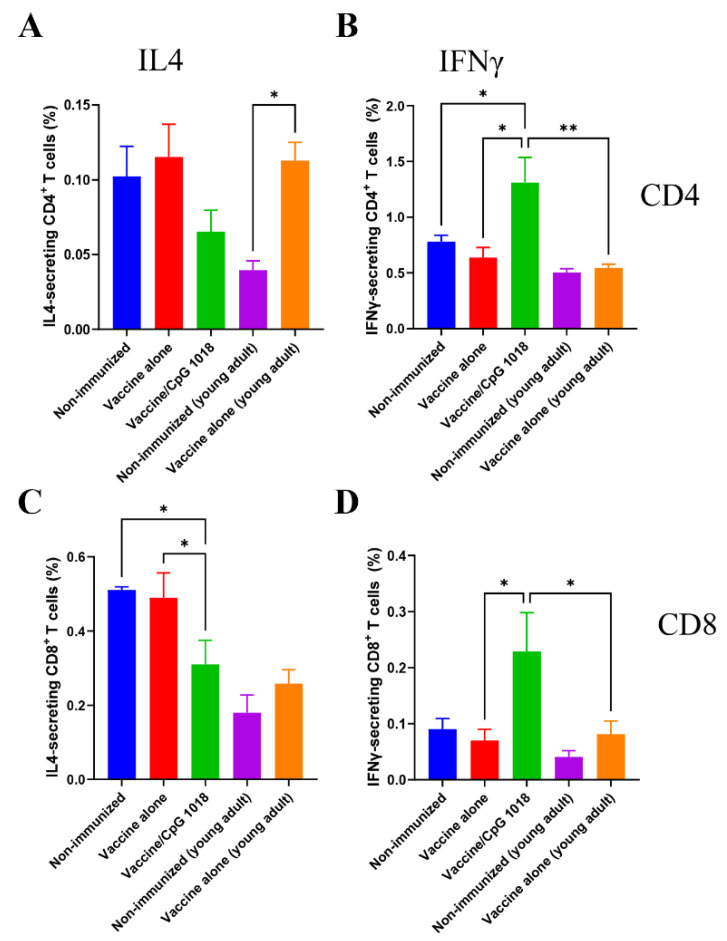
Quantitative analysis of cytokine-secreting T cells in PMBCs. The frequencies of IL4 and IFNγ-secreting cells in CD4^+^ and CD8^+^ T cells were compared between groups. The frequencies of IL4 and IFNγ-secreting cells in CD4^+^ T cells were shown in (**A**,**B**), respectively. The frequencies of IL4 and IFNγ-secreting cells in CD8^+^ T cells were shown in (**C**,**D**), respectively. *n* = 4–7. One-way ANOVA with Newman–Keuls multiple comparison test was used to compared differences between groups in (**A**–**D**). *, *p* < 0.05; **, *p* < 0.01.

**Figure 7 vaccines-10-01894-f007:**
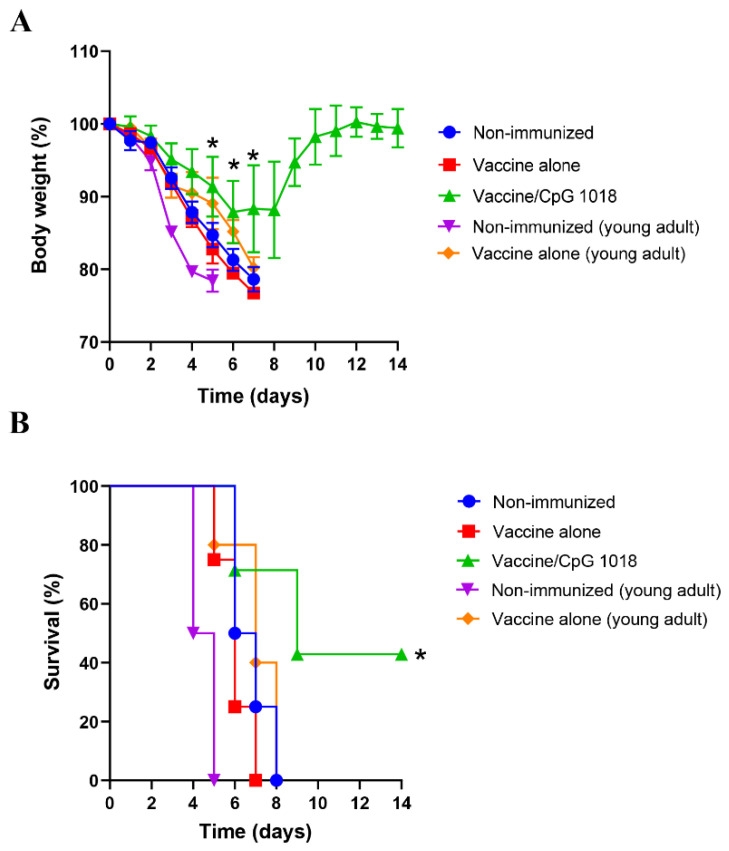
CpG 1018 enhances pdm09 vaccine-induced protection. Mice were intranasally challenged with 8× LD50 of mouse-adapted pdm09 viruses 4 weeks after immunization. Body-weight loss (**A**) and survival (**B**) were monitored daily for 14 days. Mice were regarded as dead if their body-weight loss was more than 20%. *n* = 4–7. Two-way ANOVA with Tukey’s multiple comparison test was used to compare body-weight differences between Vaccine/CpG 1018 and Vaccine alone groups in aged mice in (**A**). Log-rank test with Bonferroni correction was used to compare differences of survival between groups in (**B**). *, *p* < 0.05.

**Table 1 vaccines-10-01894-t001:** Comparison of body-weight loss between groups in young adult mice.

	Day 1	Day 2	Day 3	Day 4	Day 5
Significant level	NS	NS	NS	*p* < 0.001	*p* < 0.01

(Two-way ANOVA with Tukey’s multiple comparison test was used to compare differences between groups at different days after challenge. NS: not significant).

## Data Availability

The data that support the findings of this study are available from the corresponding author upon request.

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
