# Peer review of "Overcoming Aging-Associated Poor Influenza Vaccine Responses with CpG 1018 Adjuvant"

_vaccines, 2022, doi:10.3390/vaccines10111894_

Round 1
Reviewer 1 Report
This study help better understand the question of low efficiency of influenza vaccination in old individuals. Application of CpG 1018 adjuvant is an appropriate candidate for increasing the effectiveness of vaccination. The experimental data clearly demonstrate the activation of Th1 pathway by CpG 1018 adjuvant. The aim of this work is clear, description of both material and methods and the results are clear. Question: similar activation can be detect in other strains of mice (e.g. in BALB/c) with altered immunogenetical background as the C57BL/6? In summary: the manuscript is correctly demonstrates the usefulness of CpG 1018 adjuvant in vaccine development.
The experimental field is correctly introduced. Strengths of the manuscript are the topical theme choice, the precise experimental plan, the informative demonstration and the realistic discussion of results. However, some weaknesses are found; e.g. the study based on one inbred strain of mice, the mechanism of act of CpG 1018 adjuvant is not clarified in enough detail (better activation of antigen presenting cells compared with other adjuvants? or modification of cytokine network?).Author Response
A point-by-point response to reviewers
First of all, we would like to thank reviewers for their valuable comments and suggestions. We have thoroughly revised our manuscript either in the following point-by-point responses or in the body of the manuscript whenever appropriate. These suggestions and modifications have improved the manuscript significantly. For your convenience, the major changes have been highlighted in red in the revised manuscript.
Reviewer 1
This study help better understand the question of low efficiency of influenza vaccination in old individuals. Application of CpG 1018 adjuvant is an appropriate candidate for increasing the effectiveness of vaccination. The experimental data clearly demonstrate the activation of Th1 pathway by CpG 1018 adjuvant. The aim of this work is clear, description of both material and methods and the results are clear. Question: similar activation can be detect in other strains of mice (e.g. in BALB/c) with altered immunogenetical background as the C57BL/6? In summary: the manuscript is correctly demonstrates the usefulness of CpG 1018 adjuvant in vaccine development.
Response: The major reason to use C57BL/6 species in our study is that only this species of aged mice are commercially available. Another reason to use this species is it has a good balance in humoral and cellular immune responses. BALB/c mice are often used to elicit high titers of antibody responses.
The experimental field is correctly introduced. Strengths of the manuscript are the topical theme choice, the precise experimental plan, the informative demonstration and the realistic discussion of results. However, some weaknesses are found; e.g. the study based on one inbred strain of mice, the mechanism of act of CpG 1018 adjuvant is not clarified in enough detail (better activation of antigen presenting cells compared with other adjuvants? or modification of cytokine network?).
Response: CpG 1018 adjuvant is expected to bind TLR9 and stimulate dendritic cell maturation and release of Th1 cytokines. As of such, CpG 1018 mainly enhances Th1-biased antibody responses and CTL responses. Our unpublished data indicate CpG 1018 could activate murine bone marrow-derived dendritic cells to express costimulatory molecules, such as CD40, CD80, CD86, and MHC II.
Reviewer 2 Report
Kang and colleagues have examined the effect of adding CpG to vaccination against influenza in aged mice. The flu vaccine is only moderately effective in young and middle-aged populations. It is less effective in aged populations, and developing ways to enhance the response to vaccinate in aged individuals is an important and timely topic. The authors experiments are well designed and include proper controls. I've noted several ways in which the manuscript can be improved, none of which requires additional experimentation.
Major Comments:
1. Figure legends are insufficient. Currently, the legends have only a figure titles. Legends should be revised to include key experimental details, including the number of animals in each group.
2. Figure 4: Include a pair-wise comparison between the “vaccine alone” and “vaccine/CpG 1018” groups. The main story of this paper is that CpG enhances the response over vaccine alone. Statistically significant differences should be made clear wherever p<0.05. If the difference in the responses between groups is not significant, it should be states as “not significant.” Please see major comment #1 dealing with necessary revision of figure legends.
3. Figure 5: This is a technically challenging experiment. Authors should include an additional panel in this figure that clearly explains the gating strategy. Show FSC and SSC, CD4 and CD8 populations. Authors should also briefly state their rational for choosing the gate location. Note that the percentage of IFNγ+ cells is much lower in the non-immunized CD8+ cells than the non-immunized CD4+ cells. This point is important for two reason 1) the IFNγ response in CD4+ cells from old mice is actually higher in non-immunized mice than in the vaccine alone group. 2) Most of the increase in the IFNγ response in CD4+ cells from old mice in the vaccine/CpG 1018 mice come from cells with very low levels of cytokine (contrast with the beautiful IL-4 production in cells from young mice immunized with vaccine alone). As I stated, this experiment is a technically very challenge experiment, but I worry that the authors are overstating the significance of the CpG enhancement of the Th1 response. The additional panels showing gating strategy will enable readers to draw better-informed conclusion.
4. Figure 5 and Figure 6 report data from the same experiment and should be combined into a single figure with multiple panels. To reiterate, this revised figure should include a panel detailing the gating strategy as requested in my above point.
Minor Comments:
1. Lines 164: A very minor point to be considered at the authors’ discretion, but I would not use the word “hint” here. Fig 2 shows a significantly stronger response with CpG, so I think a stronger verb would be appropriate in this sentence.
2. CpG has been previously shown not to enhance the response in aged mice, albeit via intranasal immunization (PMID 26934728). The authors should comment on the contrast between their findings and previous reports. If the authors are aware of other relevant contrasting reports, they should address those as well.
3. The colors are distracting in the dot plots and bar graphs. The groups are properly labeled, and using colors in these figures does not help communicate the results.
4. Figure 7: Increased susceptibility to challenge of non-immunized young mice as compared to non-immunized aged mice may surprise some people. I’ve intranasally challenged aged mice with one species of bacterial pathogen and observed the same effect. Close collaborators have performed similar experiments with other bacterial pathogens in aged mice and also seen this effect.
5. Authors should refer to “old mice” as “aged mice.”
